# Impact of Particulate Matter on the Exacerbation of Immunoglobulin A Nephropathy: An Animal Experimental Study

**DOI:** 10.3390/ijms26199387

**Published:** 2025-09-25

**Authors:** Minhyeok Lee, Yeon Woo Lee, Daeun Kang, Ji Woong Son, Wan Jin Hwang, Sin Yung Woo, Mi Jin Hong, Yusin Pak, Se-Hee Yoon, Won Min Hwang, Sung-Ro Yun, Yohan Park

**Affiliations:** 1Myunggok Medical Research Institute, College of Medicine, Konyang University, Daejeon 35365, Republic of Korea; spacetravel@naver.com (M.L.); kigasuk@gmail.com (Y.W.L.); 2Division of Pulmonology, Department of Internal Medicine, Konyang University Hospital, College of Medicine, Konyang University, Daejeon 35365, Republic of Koreask1609@kyuh.ac.kr (J.W.S.); 3Department of Thoracic and Cardiovascular Surgery, Konyang University Hospital, College of Medicine, Konyang University, Daejeon 35365, Republic of Korea; 200687@kyuh.ac.kr; 4Division of Endocrinology, Department of Internal Medicine, Konyang University Hospital, College of Medicine, Konyang University, Daejeon 35365, Republic of Korea; 200727@kyuh.ac.kr; 5Department of Rehabilitation, Konyang University Hospital, College of Medicine, Konyang University, Daejeon 35365, Republic of Korea; 6Sensor System Research Center, Korea Institute of Science and Technology (KIST), Seoul 02792, Republic of Korea; yusinpak@kist.re.kr; 7Division of Nephrology, Department of Internal Medicine, Konyang University Hospital, College of Medicine, Konyang University, Daejeon 35365, Republic of Korea; sehei@hanmail.net (S.-H.Y.); hwangwm@kyuh.ac.kr (W.M.H.); sryun@kyuh.ac.kr (S.-R.Y.)

**Keywords:** IgA nephropathy, particulate matter, Toll-like receptor 9, APRIL protein, B-lymphocytes

## Abstract

Particulate matter (PM) exposure is linked to chronic kidney disease; however, its effect on immunoglobulin A (IgA) nephropathy (IgAN) remains unclear. We investigated whether PM exposure exacerbates IgAN in a mouse model. HIGA mice (IgAN model) and BALB/c controls were exposed to PM in a sealed chamber for 13 weeks. Lung Toll-like receptor 9 (TLR9) expression, serum aberrantly glycosylated IgA, A proliferation-inducing ligand (APRIL) levels, mesangial IgA deposition, and kidney pathology were assessed. RNA sequencing of splenic B cells was performed to evaluate immune-related gene expression. PM exposure increased lung TLR9 expression in both strains, particularly around pigment-laden macrophages. HIGA mice showed elevated aberrant IgA and APRIL levels, with aggravated mesangial expansion and IgA deposition. Transcriptomic analysis revealed immune dysregulation in splenic B cells of PM-exposed HIGA mice. Our findings provide experimental evidence that PM exposure aggravates IgAN via TLR9-mediated mucosal immune activation, leading to aberrant IgA glycosylation and mesangial deposition. These findings emphasize that reducing PM exposure may benefit patients with IgAN.

## 1. Introduction

Particulate matter (PM) in air pollution is a major health risk factor [1]. PM exerts direct inhalational toxicity on the lung and induces systemic effects that extend to multiple organs, including the cardiovascular and renal systems [2,3]. Epidemiological studies have revealed that long-term PM_2_._5_ exposure accelerates the decline in estimated glomerular filtration rate (eGFR), increases the risk of chronic kidney disease (CKD), and promotes progression to end-stage kidney disease (ESKD) [4,5,6]. Animal studies have demonstrated that PM exposure can induce chronic tissue remodeling in the kidney, such as fibrosis, mesangial expansion, and tubular deterioration, along with increased oxidative stress and inflammation [7,8,9]. These observations suggest that inflammatory activation triggered in the respiratory tract may propagate systemically and contribute to kidney injury.

Environmental nephrology has further revealed ties between airborne PM and the progression of glomerular diseases [10,11]. Recent studies demonstrated that increased ambient PM_2_._5_ levels independently elevate the risk of ESKD in immunoglobulin A (IgA) nephropathy (IgAN) patients in China [10]. Moreover, PM_2_._5_ exposure has been associated with accelerated glomerular disease progression and intrarenal inflammatory activation [12]. Population-based biopsy data from China also suggested a rising incidence of membranous nephropathy, indicating a potential environmental trend [11].

IgAN is the most common form of glomerulonephritis, and approximately 40% of cases progress to ESKD [13]. Recent advances in our understanding of the pathogenesis of IgAN have focused on the multi-hit hypothesis [14], which has driven the development of new therapeutic agents and a surge in clinical trials [15,16,17,18,19,20,21]. According to this hypothesis, galactose-deficient IgA1 (Gd-IgA1) plays a critical role in the pathogenesis of IgAN and is closely linked to mucosal immune activation [22]. Although direct evidence for PM-induced IgAN is limited, mucosal immunology provides a basis for this hypothesis [22,23]. IgAN is characterized by dysregulation of the mucosa-associated lymphoid system, and recent reviews have reported that aberrant mucosa-associated lymphoid tissue (MALT) responses promote Gd-IgA1 production and immune complex formation, which are key drivers of disease pathogenesis [14,24]. Since exposure to PM occurs predominantly through the respiratory tract, where it induces inflammatory responses in the respiratory mucosae [25], it may exacerbate IgAN.

Indeed, epidemiologic evidence has suggested that regional differences in PM_2.5_ exposure may influence IgAN progression [10]. Despite the growing recognition of PM as a critical environmental and health issue, the potential mechanistic link between PM exposure and IgAN exacerbation has not been widely investigated. To address this gap, we experimentally investigated whether PM exposure aggravates IgAN using a spontaneous IgAN mouse model.

## 2. Results

### 2.1. Toll-Like Receptor 9 Expression in Lung Tissues Following PM Exposure

Enzyme-linked immunosorbent assay (ELISA) results revealed that Toll-like receptor 9 (TLR9) expression in the left lung of the PM-exposed (E) groups (BALB/c-E and HIGA-E) was significantly increased compared with that in the PM non-exposed (NE) groups (BALB/c-NE and HIGA-NE; Figure 1a). Hematoxylin and eosin (H&E) staining of lung tissues from BALB/c and HIGA mice revealed pigmented macrophages in E groups (BALB/c-E and HIGA-E) and an increased number of surrounding inflammatory cells. Furthermore, immunohistochemistry (IHC) staining revealed TLR9 expression in the pigmented macrophages (Figure 1b).

### 2.2. Comparison of Serum IgA, A Proliferation-Inducing Ligand, and Glycosylated IgA Concentrations

Serum IgA levels were significantly higher in the HIGA groups (6069 ± 493 μg/mL and 6743 ± 758 μg/mL in the HIGA-NE and HIGA-E groups, respectively) than in the BALB/c groups (699 ± 69 μg/mL and 644 ± 70 μg/mL in the BALB/c-NE and BALB/c-E groups, respectively; Figure 2a). Similarly, the serum A Proliferation-Inducing Ligand (APRIL) levels were notably higher in the HIGA groups (3.75 ± 0.75 ng/mL and 5.04 ± 2.27 ng/mL in the HIGA-NE and HIGA-E groups, respectively) than in the BALB/c groups (1.42 ± 0.70 ng/mL and 1.35 ± 0.32 ng/mL in the BALB/c-NE and BALB/c-E groups, respectively), with a tendency for the highest levels to occur in the HIGA-E group (Figure 2b).

Moreover, serum IgA in the HIGA-E group exhibited reduced reactivity to the lectins *Ricinus communis* Agglutinin I (RCA-I; 1.32 ± 0.10 at optical density [OD] 450 nm in the HIGA-E group vs. 1.75 ± 0.31 at OD 450 nm in the HIGA-NE group) and Sambucus Nigra Lectin (SNL; 1.30 ± 0.15 at OD 450 nm in the HIGA-E group vs. 1.73 ± 0.28 at OD 450 nm in the HIGA-NE group), indicating lower galactose and sialic acid contents in the N-glycans of IgA from the HIGA-E group (Figure 2c,d).

### 2.3. Comparison of Kidney Histopathology, Transmission Electron Microscopy, and IHC Staining Findings

Periodic acid–Schiff (PAS) staining of the kidney tissues revealed no significant changes in the glomeruli among the BALB/c groups, whereas pronounced mesangial expansion and mesangial cell proliferation were observed among the HIGA groups. Notably, frequent occurrences of glomerular sclerosis were observed in the HIGA-E group. The renal injury score was highest in the HIGA-E group, followed by the HIGA-NE group (7.7 ± 0.6, 5.3 ± 0.6, 3.0 ± 0.0, and 2.0 ± 0.0 in the HIGA-E, HIGA-NE, BALB/c-E, and BALB/c-NE groups, respectively; Figure 3a). Transmission Electron Microscopy (TEM) revealed no evidence of electron-dense deposits in the mesangium of the BALB/c groups. In contrast, mesangial electron-dense deposits were identified in the HIGA groups (Figure 3b).

IHC of the kidney tissues revealed no significant IgA deposition in the glomeruli of the BALB/c groups. However, mild mesangial IgA deposition was observed in the HIGA-NE group, and strong mesangial IgA deposition was observed in the HIGA-E group. The IgA staining area fraction per glomerulus was 21.9% in the HIGA-NE mice and significantly greater (40.4%) in the HIGA-E mice (Figure 4).

### 2.4. Comparison of Splenic B Cell Gene Expression According to Mouse Strain and PM Exposure

At baseline, extensive gene expression differences were observed between HIGA and BALB/c mice, reflecting their divergent genetic backgrounds and IgA phenotypes (501 genes upregulated and 357 downregulated in HIGA compared with BALB/c). Importantly, PM exposure induced negligible transcriptional changes in BALB/c mice (only 2 genes altered), whereas 37 genes were significantly downregulated in HIGA mice, underscoring the heightened susceptibility of the HIGA strain to PM-induced immune perturbation (Figure 5a).

Principal component analysis (PCA) (Figure 5b) confirmed a clear separation by strain and partial segregation according to PM exposure. Heatmaps of representative differentially expressed genes (DEGs) further illustrated distinct gene expression signatures between strains (Figure 5c) and within HIGA mice after PM exposure (Figure 5d).

Functional enrichment analysis revealed that the DEGs in HIGA mice were predominantly associated with immune regulation, chromatin remodeling, and nucleosome assembly, suggesting impaired immune responses and altered genomic stability following PM exposure. Detailed Kyoto Encyclopedia of Genes and Genomes (KEGG) and Gene Ontology (GO) enrichment results are provided in Appendix A.

## 3. Discussion

Our findings revealed that PM exposure exacerbates the pathogenesis of IgAN, due to increased TLR9 expression, aberrant IgA glycosylation, and enhanced IgA deposition in the kidneys. The findings show the differential susceptibility of genetically predisposed HIGA mice compared with BALB/c mice, highlighting the interplay between environmental factors and genetic predisposition in the progression of IgAN.

HIGA mice, used as the IgAN disease model, were developed by interbreeding animals with high IgA levels within the ddY strain, which spontaneously develops IgAN [26]. HIGA mice are characterized by remarkably elevated serum IgA levels, measured in this study at 6000–7000 μg/mL, more than 10 times higher than those observed in BALB/c mice. These findings align with the data provided by the supplier (9482 ± 1952 μg/mL in the HIGA mouse and 1345 ± 769 μg/mL in the BALB/c mouse, as reported by Japan SLC, Inc.) [27]. IgAN is closely associated with mucosal immunity, and recent studies have reported that stimulation with CpG oligonucleotides, a TLR9 ligand, induces Interleukin-6 (IL-6) and APRIL production, leading to the overproduction of aberrantly glycosylated IgA [28]. Our findings demonstrated that PM exposure upregulated TLR9 expression in HIGA and BALB/c mice. In line with a previous study, only HIGA mice exhibited significant increases in serum APRIL and aberrantly glycosylated IgA levels. Furthermore, histopathological analysis revealed the highest renal injury scores and most pronounced mesangial IgA depositions in the HIGA-E group. These results support the hypothesis that PM exposure induces mucosal immune activation in IgAN patients, promoting aberrant IgA overproduction, which ultimately leads to kidney damage [14,22].

This study also demonstrated that PM exposure significantly increases TLR9 expression in lung tissues, particularly in pigment-laden macrophages, alongside the accumulation of inflammatory cells. The observed increase in TLR9 expression in surrounding inflammatory cells associated with PM phagocytosis by lung macrophages is particularly intriguing. The effects of air pollutants on mucosal immunity in the lower respiratory tract are well-documented. These pollutants influence innate immunity by activating various TLR and NOD-like receptor signaling pathways [25]. In mouse models exposed to smoking, TLR9 signaling plays a crucial role in the development of lung diseases, such as emphysema [29,30]. The role of mucosal immunity in IgAN pathogenesis is also well-established, with TLRs being key mediators [28,31]. Although previous research has focused on upper respiratory tract mucosal immunity [22], this study has emphasized that lung tissue mucosal immunity in the lower respiratory tract can also contribute to IgAN pathogenesis.

Gene expression analysis of CD19+ splenic B cells revealed significant differences between HIGA and BALB/c mice. These findings are consistent with those of previous studies investigating the relationship between B cells and the development of IgAN [32]. This study revealed alterations in various signaling pathways involved in inflammation in CD19+ splenic B cells, providing further evidence of their role in IgAN pathogenesis [32]. Furthermore, HIGA mice showed greater changes in gene expression in response to PM exposure than BALB/c mice, indicating that HIGA mice have greater immunological vulnerability to environmental stress. In addition, RNA-seq analysis revealed suppression of immune-related and chromatin assembly pathways in splenic B cells from HIGA mice after PM exposure. This observation is consistent with recent evidence that PM_2.5_ alters chromatin dynamics and epigenetic regulation, including nucleosome assembly and histone modification [33,34,35]. Such changes may impair B-cell immunoregulation and contribute to epigenetic instability under PM stress.

This study has several limitations. First, HIGA mice were used as the IgAN disease model. Although these mice consistently exhibit high serum IgA levels, these levels are not correlated with the kidney pathology of IgAN [36]. Furthermore, the onset of IgAN in HIGA mice is inconsistent, leading to variability in the severity of kidney pathology at specific time points for each mouse, which may result in heterogeneous findings [36]. Second, in the lectin-binding assays, RCA-I showed that glycosylated IgA levels were significantly lower in the HIGA-E group. In contrast, SNL tended to have lower glycosylated IgA levels, but the results were not statistically significant. Moreover, although mesangial IgA deposition was observed in the kidney tissue of the HIGA groups, C3 deposition was not evident. This limitation of HIGA mice was also highlighted in a previous study, which found that although HIGA mice exhibit high serum IgA levels, they do not consistently produce aberrantly glycosylated IgA [37]. Furthermore, their kidney pathology showed prominent mesangial IgA deposition but lacked consistent C3 deposition; consequently, their findings did not fully align with those for human IgAN [37]. However, a previous study on patients with IgAN reported that 18.1% of patients were C3 negative, indicating that the absence of C3 deposition does not necessarily preclude disease manifestation or progression [38]. To address these limitations, employing early-onset ddY mice may be beneficial, as they have recently been introduced as a more reliable model of spontaneous IgAN [14,37]. However, these mice are not yet commercially available. Despite the inherent limitations of the HIGA mouse model, the findings of this study—particularly, the consistently higher IgAN severity observed in the HIGA-E group compared with the HIGA-NE group, along with changes consistent with the pathogenesis of IgAN and stronger gene expression changes in splenic B cells in response to PM exposure compared with the control in the disease model—provide experimental evidence that PM exposure may exacerbate IgAN. Third, this study was limited by the small number of mice used. Although the fundamental limitations of the HIGA mouse model were inevitable, experiments with a larger sample size may have provided more consistent and robust evidence supporting the hypothesis. Due to resource constraints, this study included only three mice per group. Future studies with sufficient early-onset ddY mice could provide more substantial experimental evidence. Fourth, RNA-seq analysis revealed differential gene expression; however, we were unable to perform functional validation. This limitation reflects the lack of commercially available knockout mouse models on the HIGA background and the absence of a robust in vitro cellular system for IgAN. Future studies should therefore incorporate complementary approaches, such as targeted knockdown or blocking of candidate pathways in related mouse strains, or the use of emerging humanized B cell culture systems, to determine whether the observed transcriptional changes directly contribute to disease progression. Finally, we could not determine the tissue origin of APRIL expression. Previous studies have indicated that APRIL is predominantly produced by myeloid cells, such as monocytes, macrophages, and dendritic cells, as well as by activated lymphocytes [39,40]. Clarifying whether splenic B cells, lung myeloid cells, or other immune cell subsets predominantly contribute to APRIL production would provide mechanistic insights. In addition, due to limited sample availability, we were unable to measure serum blood urea nitrogen and creatinine or proteinuria, which are clinically relevant markers of renal function in IgAN.

Despite these limitations, this study is significant as it is the first experimental research to suggest that PM exposure may exacerbate IgAN. Many previous studies analyzing PM toxicity have used intratracheal instillation as a PM exposure method. However, recent reports indicate that PM toxicity observed with natural inhalation is less pronounced compared with that observed with intratracheal instillation [41]. Nevertheless, no doubt exists that natural inhalation more closely resembles the real-world environment. This study designed and developed a sealed PM exposure cage to facilitate controlled environment experiments. Using an automated PM measurement and exposure program (Appendix A), the target PM concentrations (PM_2.5_ and PM_10_) in the air of the sealed cage were reliably maintained. Observing IgAN exacerbation in a spontaneous IgAN mouse model using this method underscores the significance of this study, as it suggests that PM exposure in a real-world environment could contribute to disease progression in IgAN patients.

This study demonstrated that PM exposure negatively affects IgAN through increased TLR9 expression, aberrant IgA glycosylation, and increased IgA deposition in the kidneys. The findings revealed greater susceptibility of genetically predisposed HIGA mice than BALB/c mice, highlighting the impacts of environmental factors and genetic predisposition on the progression of IgAN. These results present pathological evidence of IgAN exacerbation, highlighting the combined impact of environmental exposure and genetic predisposition.

## 4. Materials and Methods

### 4.1. Animal Experiment and PM Exposure

All experimental procedures were approved by the Institutional Animal Care and Use Committee (IACUC) of Konyang University (Approval No: P-23-11-E-01, 21 April 2023) and were conducted under the institutional guidelines and regulations. A brief overview of the experiment is presented in Figure 6. Ten-week-old female BALB/c mice (*n* = 6) and HIGA mice (*n* = 6; a model for IgAN) were purchased from Japan SLC, Inc. (Shizuoka, Japan). Based on PM exposure, groups (*n* = 3) were classified as BALB/c-NE, BALB/c-E, HIGA-NE, and HIGA-E. The PM used for exposure was ISO 12103-1, A1 Ultrafine Test Dust (Powder Technology Inc., USA), a standardized reference material derived from Arizona desert sand, specifically sourced from the Salt River area in Arizona. Detailed information on the chemical composition of the dust is provided in the Appendix A. The PM exposure chamber (40 × 20 × 25 cm; Appendix A) was constructed from a plastic enclosure with silicone tubing installed on the bottom. To continuously monitor PM concentrations, a PM sensor (PMS5003ST, Plantower Technology, Nanchang, China) was positioned at the center of the chamber.

After a one-week acclimation period, the experimental group was exposed to PM for 1 h daily, 5 days a week, for 13 weeks. In contrast, the control group was exposed to air filtered through a PM_2.5_ filter for the same duration. The concentrations of PM_2.5_ and PM_10_ in the exposure and control groups are presented in Appendix A. After exposure to the experimental environment for 13 weeks, mice were humanely euthanized at 24 weeks of age, and blood, spleen, lung, and kidney samples were collected for subsequent analysis.

### 4.2. Measurement of TLR9 Levels in Lung Tissues

Lung TLR9 levels were measured using an ELISA kit (Antibodies, Cat. No. A77428). The left lung of each mouse was homogenized, and the supernatant was collected for the assay. Samples were diluted 1:60, and the assay was performed according to the manufacturer’s protocol, using a standard curve with 1 ng/mL TLR9. The optical density was measured at 450 nm.

### 4.3. Serum IgA, Glycosylated IgA, and APRIL Measurements

Serum samples were obtained from blood samples. Serum IgA levels were measured using an ELISA kit (Bethyl Laboratories, Cat. No. E99-103). Aberrantly glycosylated IgA was analyzed using two specific lectins: SNL (Vector Laboratories, Cat. No. B-1305) for detecting terminal sialic acid residues and RCA-I (Vector Laboratories, Cat. No. B-1085) for identifying terminal galactose residues. This analysis utilized the same IgA ELISA kit, with SNL or RCA-I added to the respective wells following sample loading, and detection was achieved using an avidin-horseradish peroxidase conjugate. The serum levels of APRIL were measured using a specific ELISA kit (LSBio, Cat. No. LS-F12734), according to the manufacturer’s protocol.

### 4.4. Histopathology Analyses

Sections were stained with Harris hematoxylin, differentiated with 1% HCl, and treated with 1% ammonia water to obtain blue nuclei. After staining with alcoholic eosin, sections were dehydrated, cleared, and mounted for analysis. PAS staining of the kidney was performed by deparaffinizing and rehydrating tissue sections, followed by treatment with 0.1% periodic acid and staining with Schiff’s reagent. Sections were counterstained with Mayer’s hematoxylin, then dehydrated, cleared, and mounted for analysis. IHC staining was performed using TLR9 antibody (Abcam, Anti-TLR9 antibody, ab134368) for lung tissues and IgA antibody (Abcam, Goat Anti-Mouse IgA alpha chain [FITC], ab97234) for kidney tissues.

The renal injury score was determined by quantitatively analyzing glomeruli for segmental and global sclerosis, mesangial cell proliferation, and mesangial matrix expansion. Each section was graded semi-quantitatively as follows: 0 (0%), 1 (1–24%), 2 (25–49%), and 3 (>50% of total glomeruli), with a maximum possible score of 9 per section [30]. A mean of 76.3 ± 18.7 glomeruli per kidney was observed.

TEM was performed on the kidney tissues. Samples were fixed in 2.5% glutaraldehyde in cacodylate buffer, post-fixed with 2% osmium tetroxide, and stained with 2% uranyl acetate. After dehydration using a graded ethanol series and acetone, samples were infiltrated with resin mixtures and polymerized at 60 °C for 72 h. Ultrathin sections (50–70 nm) were collected on copper grids, post-stained with uranyl acetate and lead citrate, and examined using TEM.

### 4.5. Isolation and Culture of Splenic B Cells and RNA Sequencing Analysis

Splenic B cells were isolated via magnetic-activated cell sorting (MACS) according to the manufacturer’s protocol, and purity was validated by fluorescence-activated cell sorting (FACS; Appendix A). Total RNA was extracted, and RNA integrity was confirmed prior to sequencing. RNA sequencing was performed on an Illumina platform, and raw reads were subjected to quality control, adaptor trimming, and alignment to the mouse reference genome (GRCm39).

Raw read counts were processed using the iDEP 96 web-based platform [42]. Genes with low counts were filtered using a minimum threshold of counts per million (CPM > 0.5 in at least one sample). Count data were transformed by a variance stabilizing transformation (VST) for exploratory analyses, including PCA. Differential expression analysis was conducted using DESeq2 with a minimum fold-change cutoff of 2 and an adjusted *p*-value (Benjamini–Hochberg false discovery rate [FDR] < 0.1), and PCA plots were generated within iDEP.

Functional enrichment analysis was performed using GO (Biological Process) and KEGG annotations implemented in iDEP 96. DEG sets defined by FDR < 0.1 and |log_2_FC| ≥ 1 were used as input. Enrichment analysis was conducted using the filtered background gene list, and significance was determined by Benjamini–Hochberg adjusted *p*-values. Pathways with adjusted *p* < 0.05 were considered significantly enriched. Heatmaps of the top 15 differentially expressed genes were generated using the SRplot web tool, with expression values row-scaled and clustered by complete linkage with Euclidean distance [43]. Graphical visualization of enrichment results was also refined using SRplot [43].

### 4.6. Statistical Analysis

All statistical analyses were conducted using SPSS version 17.0 (IBM Corp., Chicago, IL, USA) and GraphPad Prism version 6.0 (GraphPad Software, La Jolla, CA, USA). The data are presented as the mean with individual values, and comparisons between groups were performed using the Mann–Whitney U test. A *p*-value ≤ 0.05 was considered statistically significant.

## Figures and Tables

**Figure 1 ijms-26-09387-f001:**
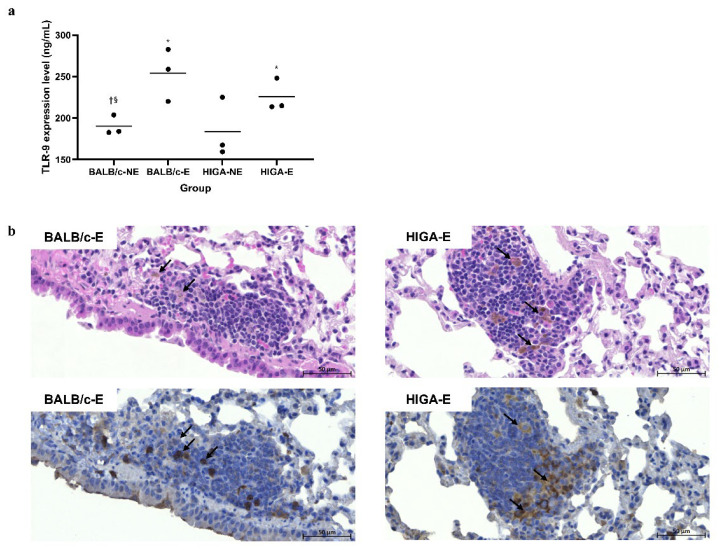
**Comparison of pathology and enzyme-linked immunosorbent assay results for Toll-like receptor 9 expression in representative lung tissue.** (**a**) Toll-like receptor 9 (TLR9) expression levels in the left lung were measured by enzyme-linked immunosorbent assay (ELISA). TLR9 expression was significantly increased in the particulate matter (PM)-exposed (E) groups (BALB/c-E and HIGA-E) compared with the non-exposed (NE) groups (BALB/c-NE and HIGA-NE). Data are presented as the mean and each specific value. Statistical significance: * *p* < 0.05 compared with the BALB/c-NE group; ^†^ *p* < 0.05 compared with the BALB/c-E group; ^§^ *p* < 0.05 compared with the HIGA-E group. (**b**) Representative histopathological images of right lung tissues from BALB/c-E and HIGA-E mice. Hematoxylin and eosin (H&E) staining (top row) reveals the presence of pigmented macrophages and increased infiltration of surrounding inflammatory cells (arrows) in the PM-E groups. Immunohistochemistry (IHC) for TLR9 (bottom row) demonstrates elevated TLR9 expression, particularly around the pigmented macrophages (arrows) in the BALB/c-E and HIGA-E groups. Scale bars: 50 μm.

**Figure 2 ijms-26-09387-f002:**
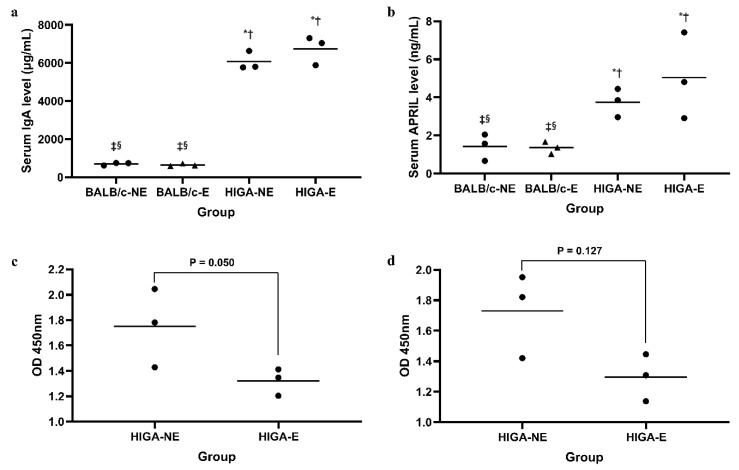
**Comparison of serum IgA, glycosylated IgA, and APRIL concentrations.** (**a**) Serum IgA levels measured by ELISA. IgA levels were significantly higher in the HIGA groups compared with the BALB/c groups. (**b**) Serum APRIL levels measured by ELISA. APRIL levels were markedly elevated in the HIGA groups, with the highest levels observed in the HIGA-E group. (**c**,**d**) Glycosylated IgA levels were analyzed using *Ricinus communis* Agglutinin I (RCA-I) and Sambucus Nigra Lectin (SNL). The HIGA-E group exhibited reduced reactivity to both lectins, indicating lower galactose and sialic acid content in IgA N-glycans. In (**a**,**b**), * *p* < 0.05 compared with the BALB/c-NE group; ^†^ *p* < 0.05 compared with the BALB/c-E group; ^‡^ *p* < 0.05 compared with the HIGA-NE group; ^§^ *p* < 0.05 compared with the HIGA-E group.

**Figure 3 ijms-26-09387-f003:**
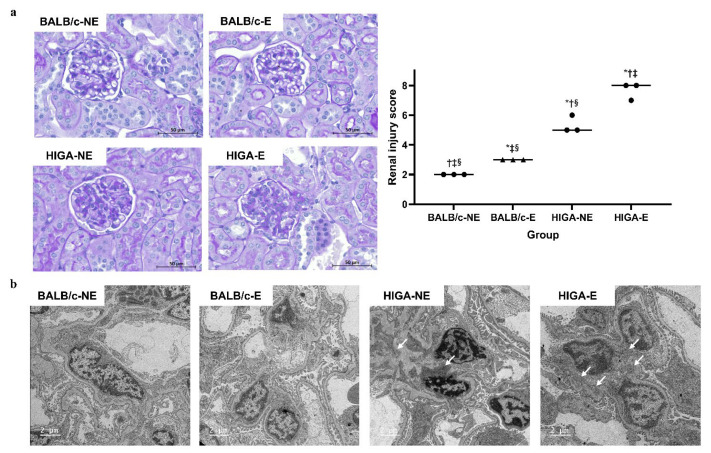
**Comparison of pathology and renal injury score in kidney tissue.** (**a**) Periodic acid–Schiff (PAS)-stained kidney tissues from BALB/c and HIGA groups. The BALB/c groups showed no significant glomerular changes, whereas the HIGA groups displayed marked mesangial expansion and mesangial cell proliferation. Glomerular sclerosis was frequently observed in the HIGA-E group, which had the highest renal injury score, followed by the HIGA-NE group. * *p* < 0.05 compared with the BALB/c-NE group; ^†^
*p* < 0.05 compared with the BALB/c-E group; ^‡^
*p* < 0.05 compared with the HIGA-NE group; ^§^
*p* < 0.05 compared with the HIGA-E group. Scale bars: 50 μm. (**b**) Transmission Electron Microscopy (TEM) images of glomeruli. No mesangial electron-dense deposits were observed in the BALB/c groups; however, they were observed in the HIGA groups (arrows). Scale bars: 2 μm.

**Figure 4 ijms-26-09387-f004:**
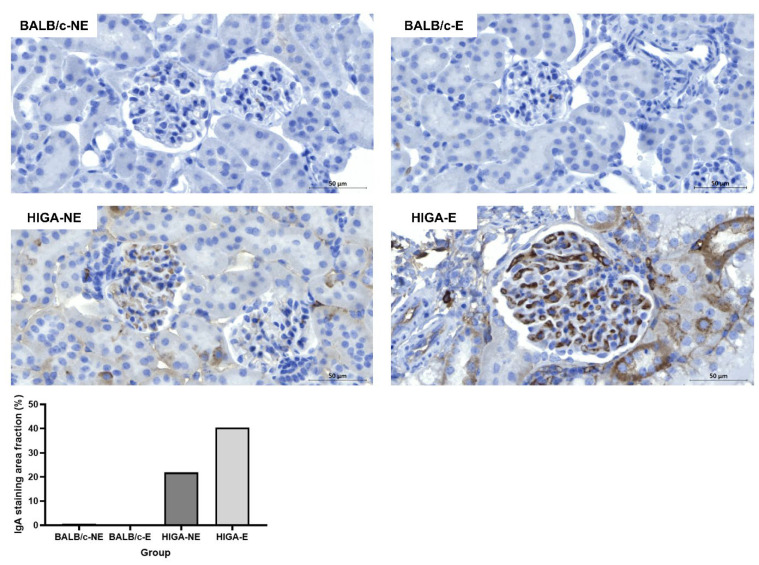
**Comparison of IgA IHC results in kidney tissue**. Representative IHC images of IgA staining in kidney tissues from BALB/c and HIGA groups. The BALB/c groups (BALB/c-NE and BALB/c-E) showed no significant IgA deposition in the glomeruli. In contrast, the HIGA-NE group exhibited mild mesangial IgA deposition, whereas the HIGA-E group displayed intense and extensive mesangial IgA deposition. Quantitative analysis of the IgA staining area fraction per glomerulus shows a higher fraction in the HIGA-E group than in the other groups. Scale bars: 50 μm.

**Figure 5 ijms-26-09387-f005:**
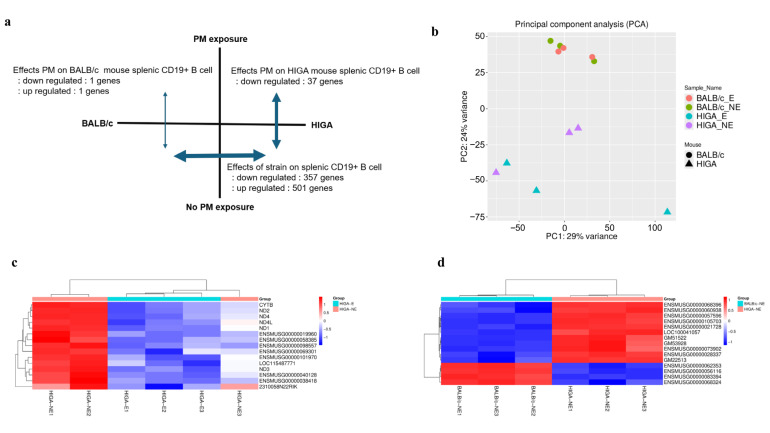
**Transcriptomic alterations in splenic CD19^+^ B cells following PM exposure.** (**a**) Summary of differentially expressed genes (DEGs) in splenic CD19^+^ B cells. At baseline, HIGA and BALB/c mice exhibited substantial gene expression differences, consistent with their distinct genetic backgrounds and IgA phenotypes. PM exposure exerted minimal impact on BALB/c mice (2 genes), but significantly downregulated 37 genes in HIGA mice, highlighting their greater susceptibility to PM. (**b**) Principal component analysis of CD19^+^ B cell transcriptomes showing clear separation by strain and partial segregation by PM exposure (*n* = 3 per group). (**c**) Heatmap of the top 15 DEGs between HIGA-E and HIGA-NE groups. (**d**) Heatmap of the top 15 DEGs between HIGA-NE and BALB/c-NE groups. Red indicates higher relative expression, and blue indicates lower relative expression.

**Figure 6 ijms-26-09387-f006:**
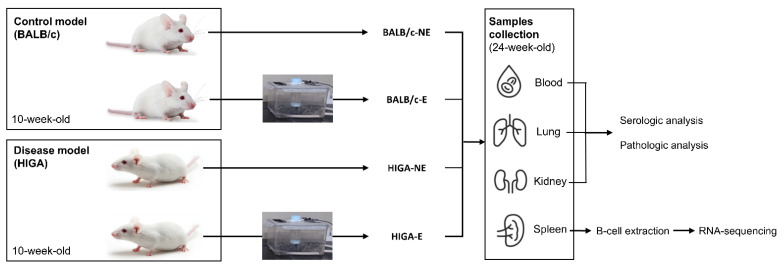
**Study design**. Overview of the experimental setup. The study involved control (BALB/c) and disease model (HIGA) mice, all of which were 10 weeks of age at the start of the experiment. Each group was divided into two subgroups: NE and E to PM. After a one-week acclimation period, PM exposure occurred for 1 h daily, 5 days per week, for 13 weeks. At 24 weeks of age, mice were euthanized, and blood, lungs, kidneys, and spleen were collected for analysis. Blood, lung, and kidney samples were analyzed for serologic and pathological changes, while spleens were processed for B cell extraction and RNA-seq to assess differential gene expression.

## Data Availability

The raw data supporting the conclusions of this article will be made available by the authors upon request.

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
