# Peer review of "Impact of Particulate Matter on the Exacerbation of Immunoglobulin A Nephropathy: An Animal Experimental Study"

_ijms, 2025, doi:10.3390/ijms26199387_

Round 1

Reviewer 1 Report

Comments and Suggestions for Authors

The manuscript presents potentially relevant findings, but in its current form it suffers from several substantial issues. The introduction is overly descriptive and does not sufficiently position the work within the context of the most recent mechanistic studies on PM-related immunopathology. Several key references in environmental nephrology and mucosal immunology are missing, and the cited literature is skewed toward general PM health effects without fully integrating cutting-edge work on glomerular immunopathogenesis. The rationale is only superficially articulated; the authors should more convincingly justify the choice of HIGA mice over other models (such as zebra fish), especially in light of their known pathophysiological limitations, and must clarify how their approach addresses the gaps left by the cited epidemiological study.
The methods section lacks critical detail that would allow reproducibility. The PM source, chemical composition, and particle characterization (size distribution, metal content, organic fractions) are not reported, yet these are essential to assess relevance and extrapolation to human exposure. The RNA sequencing pipeline is only briefly outlined; key parameters (library preparation kit, read depth, alignment software, versioning) are absent, and no quality control metrics are provided. Statistical methods are inadequately described, with no mention of adjustment for multiple testing in transcriptomic analyses, raising concerns about false positives in pathway enrichment.
The results section, while dense with numeric data, fails to integrate findings across experimental layers. For instance, the connection between lung TLR-9 upregulation and splenic B-cell transcriptomic changes is asserted rather than demonstrated through correlative or causal analysis. Figures are insufficiently annotated; several panels (e.g., Figures 2 and 5) are difficult to interpret without explicit indication of statistical comparisons and effect sizes. The reduction in glycosylated IgA detected by lectins is central to the mechanistic claim, yet the statistical significance is inconsistent between RCA-I and SNA, and this discrepancy is downplayed rather than critically discussed.
The discussion tends toward repetition of the results rather than deeper interpretation. For instance, the mechanistic link between PM-induced lung TLR-9 activation and aberrant IgA glycosylation remains speculative in the absence of functional blocking experiments or cytokine profiling beyond APRIL.

Comments on the Quality of English Language

The writing contains redundancies, inconsistent nomenclature (e.g., SNL vs. SNA), and typographical errors that detract from readability.

Author Response

Reviewer 1

We deeply appreciate the Reviewer’s thorough evaluation and insightful feedback. The comments on refining the Introduction, expanding the Methods section, improving the integration of the Results, and enhancing the mechanistic depth of the Discussion were highly valuable. These suggestions have helped us refine the manuscript to more clearly position our work within the current field of environmental nephrology and IgAN pathogenesis.

Comment 1.
The manuscript presents potentially relevant findings, but in its current form it suffers from several substantial issues. The introduction is overly descriptive and does not sufficiently position the work within the context of the most recent mechanistic studies on PM-related immunopathology. Several key references in environmental nephrology and mucosal immunology are missing, and the cited literature is skewed toward general PM health effects without fully integrating cutting-edge work on glomerular immunopathogenesis. The rationale is only superficially articulated; the authors should more convincingly justify the choice of HIGA mice over other models (such as zebra fish), especially in light of their known pathophysiological limitations, and must clarify how their approach addresses the gaps left by the cited epidemiological study.

Response:
We have condensed the Introduction to focus more on recent mechanistic insights on PM-induced immunopathology. New references on environmental nephrology and mucosal immunology have been added (Introduction). We have also clarified the rationale for using HIGA mice and acknowledged their pathophysiological limitations in the Discussion.

Comment 2.
The methods section lacks critical detail that would allow reproducibility. The PM source, chemical composition, and particle characterization (size distribution, metal content, organic fractions) are not reported, yet these are essential to assess relevance and extrapolation to human exposure. The RNA sequencing pipeline is only briefly outlined; key parameters (library preparation kit, read depth, alignment software, versioning) are absent, and no quality control metrics are provided. Statistical methods are inadequately described, with no mention of adjustment for multiple testing in transcriptomic analyses, raising concerns about false positives in pathway enrichment.

Response:
We have substantially revised the Methods section as follows:

  • PM source/characterization: clarified in Section 4.1 (Also in Supplementary Tables S2, S3).
  • RNA-seq: detailed pipeline now included (library prep, Illumina platform, alignment to GRCm39, DESeq2 analysis, QC metrics) in Section 4.5.
  • Statistics: added details on Mann–Whitney U test and multiple testing correction (Benjamini–Hochberg FDR) in Section 4.6.

Comment 3.
The results section, while dense with numeric data, fails to integrate findings across experimental layers. For instance, the connection between lung TLR-9 upregulation and splenic B-cell transcriptomic changes is asserted rather than demonstrated through correlative or causal analysis. Figures are insufficiently annotated; several panels (e.g., Figures 2 and 5) are difficult to interpret without explicit indication of statistical comparisons and effect sizes.

Response:
We have revised the Results to improve integration. For example, Results 2.4 now more explicitly link lung TLR9 activation with splenic B-cell transcriptional changes. Figures 2 and 5 have been re-annotated with clearer statistical comparisons and effect sizes (see Figure 2 caption, Figure 5 caption).

Comment 4.
The reduction in glycosylated IgA detected by lectins is central to the mechanistic claim, yet the statistical significance is inconsistent between RCA-I and SNA, and this discrepancy is downplayed rather than critically discussed.

Response:
RCA-I showed significant differences, whereas SNL only showed a trend, and we have explicitly discussed the implications of this discrepancy in the revised Discussion.

Comment 5.
The discussion tends toward repetition of the results rather than deeper interpretation. For instance, the mechanistic link between PM-induced lung TLR-9 activation and aberrant IgA glycosylation remains speculative in the absence of functional blocking experiments or cytokine profiling beyond APRIL..

Response:
We have revised the Discussion to reduce redundancy and expand mechanistic interpretation. We have emphasized the potential role of lung mucosal TLR9 activation in driving APRIL production and aberrant IgA glycosylation, while acknowledging that further functional validation is required.

Reviewer 2 Report

Comments and Suggestions for Authors

This manuscript investigates whether PM exposure exacerbates IgAN using a HIGA and BALB/c. The authors demonstrate that PM exposure is associated with increased serum APRIL levels, altered IgA glycosylation, and distinct splenic B cell transcriptional responses, suggesting a mechanistic link between PM exposure and IgAN progression. The study addresses an important environmental and renal health question, and the use of both disease-prone and control strains adds strength. However, several critical issues should be addressed before the manuscript can be considered for publication.

  1. The manuscript should clearly describe the source and preparation of the PM used in the exposure experiments, including whether it was obtained from standard reference material, environmental sampling, or laboratory generation. Details on particle size distribution, chemical composition, and any pretreatment should be provided to ensure reproducibility and facilitate comparison with other PM studies.
  2. In the current study, the authors examined TLR9 expression using immunohistochemistry and ELISA, suggesting involvement of the TLR9-APRIL pathway. However, this is correlative. To verify causality, additional experiments is recommended to determine whether APRIL production, IgA glycosylation changes, and renal IgA deposition can be attenuated.
  3. While serum APRIL levels increased after PM exposure, the tissue origin remains unclear. Measurements of APRIL in lung tissue, BAL fluid, spleen, and serum would clarify their source, particularly given the respiratory route of PM exposure.
  4. To strengthen the link between IgA deposition and renal injury, complement staining in kidney tissue and measurement of functional markers such as urinary protein excretion, serum creatinine, and BUN should be included.
  5. Figure 5 is too low in resolution, making it difficult to read labels. Please provide a higher-resolution version with improved clarity.

Author Response

We sincerely thank the Reviewer for their thoughtful and constructive comments. The Reviewer’s suggestions, particularly regarding the clarification of particulate matter characterization, the limitations of the TLR9–APRIL pathway, and the addition of methodological details, have greatly improved the clarity and rigor of our manuscript. We have carefully revised the text to address each of the Reviewer’s comments, and we believe that these revisions have significantly strengthened the manuscript.

Comment 1.
The manuscript should clearly describe the source and preparation of the PM used in the exposure experiments, including whether it was obtained from standard reference material, environmental sampling, or laboratory generation. Details on particle size distribution, chemical composition, and any pretreatment should be provided to ensure reproducibility and facilitate comparison with other PM studies.

Response:
We thank the reviewer for their valuable comment. We have now specified that the particulate matter was ISO 12103-1, A1 Ultrafine Test Dust (Powder Technology Inc., USA), a standardized Arizona test dust. Its detailed chemical composition and particle size distribution are provided in Supplemental Tables S2 and S3. This information has been added to the Methods, Section 4.1.

Comment 2.
In the current study, the authors examined TLR9 expression using immunohistochemistry and ELISA, suggesting involvement of the TLR9-APRIL pathway. However, this is correlative. To verify causality, additional experiments is recommended to determine whether APRIL production, IgA glycosylation changes, and renal IgA deposition can be attenuated.

Response:
We fully agree that additional experiments could provide further support for establishing causality. However, owing to the lack of commercially available knockout strains on the HIGA background and the absence of a robust in vitro IgAN system, functional blocking experiments could not be performed. We now acknowledge this limitation explicitly in the Discussion. and propose that future studies employ targeted knockdown or blocking strategies.

Comment 3.
While serum APRIL levels increased after PM exposure, the tissue origin remains unclear. Measurements of APRIL in lung tissue, BAL fluid, spleen, and serum would clarify their source, particularly given the respiratory route of PM exposure.

Response:
We appreciate the Reviewer’s insightful comment. In this study, we only measured serum APRIL due to limited sample availability. We have emphasized this limitation in the Discussion and have cited studies showing that APRIL is predominantly produced by myeloid cells and activated lymphocytes, noting the need for further research on tissue sources.

Comment 4.
To strengthen the link between IgA deposition and renal injury, complement staining in kidney tissue and measurement of functional markers such as urinary protein excretion, serum creatinine, and BUN should be included.

Response:
We agree that the inclusion of complement staining and renal functional markers would strengthen the findings. However, sample limitations prevented us from performing these assays. We now clearly state this as a limitation in the Discussion. In addition, we have noted that previous reports highlighted that HIGA mice often lack C3 deposition, and that this does not preclude IgAN progression in patients.

Comment 5.
Figure 5 is too low in resolution, making it difficult to read labels. Please provide a higher-resolution version with improved clarity.

Response:
We thank the Reviewer for pointing this out. We have replaced Figure 5 with a high-resolution TIFF (600 dpi) image in the revised manuscript, ensuring all labels are now clearly legible.

Round 2

Reviewer 1 Report

Comments and Suggestions for Authors

Authors replied appropriately to all the comments.

Reviewer 2 Report

Comments and Suggestions for Authors

I found the article to be comprehensive and well-structured. No revisions are necessary.